# Cadmium and Lead Decrease Cell–Cell Aggregation and Increase Migration and Invasion in Renca Mouse Renal Cell Carcinoma Cells

**DOI:** 10.3390/ijms20246315

**Published:** 2019-12-14

**Authors:** Ryan Akin, David Hannibal, Margaret Loida, Emily M. Stevens, Elizabeth A. Grunz-Borgmann, Alan R. Parrish

**Affiliations:** Department of Medical Pharmacology and Physiology, School of Medicine, University of Missouri, Columbia, MO 65212, USA; rdakfd@health.missouri.edu (R.A.); djhzq8@health.missouri.edu (D.H.); mllqf9@mail.missouri.edu (M.L.); emsb23@mail.missouri.edu (E.M.S.); grunze@health.missouri.edu (E.A.G.-B.)

**Keywords:** cadmium, E-cadherin, invasion, lead, matrix metalloproteinase-9, migration, p120-catenin, renal cell carcinoma

## Abstract

Metastatic renal cell carcinoma (RCC) remains an important clinical issue; the 5-year survival rate of patients with metastasis is approximately 12%, while it is 93% in those with localized disease. There is evidence that blood cadmium and lead levels are elevated in RCC. The current studies were designed to assess the impact of cadmium and lead on the progression of RCC. The disruption of homotypic cell–cell adhesion is an essential step in epithelial-to-mesenchymal transition and tumor metastasis. Therefore, we examined the impact of cadmium and lead on the cadherin/catenin complex in Renca cells—a mouse RCC cell line. Lead, but not cadmium, induced a concentration-dependent loss of E-cadherin, while cadmium, but not lead, increased p120-catenin expression, specifically isoform 1 expression. Lead also induced a substantial increase in matrix metalloproteinase-9 levels. Both cadmium and lead significantly decreased the number of Renca cell aggregates, consistent with the disruption of the cadherin/catenin complex. Both metals enhanced wound healing in a scratch assay, and increased cell migration and invasion. These data suggest that cadmium and lead promote RCC progression.

## 1. Introduction

Kidney cancer is the eighth leading cause of cancer, as well as cancer deaths [1]. It is estimated that 63,900 new cases of kidney cancer and 14,400 deaths occurred in 2017; new cases have increased by 0.7% annually over the past decade [1]. Metastatic disease is a significant clinical problem; 16% of new cases have metastasized at diagnosis, and the 5-year survival for these patients is 11.7% compared to 92.6% and 66.7% for those with localized and regional tumors, respectively [1]. Renal cell carcinoma (RCC) accounts for 80% of kidney cancers [2]. Major risk factors for RCC include excess body weight, hypertension and smoking [3]; the disease is more common in males over 60 years of age [4]. There are over 10 histological and molecular subtypes of RCC; the most common are clear cell RCC (ccRCC; 75% of cases), papillary RCC (pRCC; 10–15% of cases) and chromophobe RCC (chRCC; 5% of cases). The remaining subtypes are very rare, i.e., less than 1% incidence. In terms of initial invasion from the primary tumor, there is evidence for the crucial role of the epithelial-to-mesenchymal transition (EMT) [5,6].

Cadmium and lead are pervasive environmental contaminants; at low, chronic exposures, renal dysfunction is an important concern [7,8]. Human exposure to cadmium or lead occurs via workplace exposure or ingestion [8,9]. In addition, tobacco contains cadmium; as such, smoking is a major source of exposure [10]. A link between cadmium and renal cancer was suggested over 4 decades ago by data indicating an association between renal cancer and cadmium exposure, with a potential synergistic effect between occupational exposure and smoking [11]. A recent meta-analysis suggests that cadmium exposure is a risk factor for renal cancer [12]. With regards to lead and RCC, higher blood levels of lead are associated with RCC [13,14], and an accumulation of lead in higher stage RCC tissue has been observed (control <0.05 μg/g; RCC 1.74 μg/g) [15]. In a study of 33 non-smoking male patients with RCC, their blood levels of cadmium (control 0.027 μg/dl; RCC 0.091 μg/dl) and lead (control 0.02 μg/dl; RCC 0.09 μg/dl) were elevated 3.37- and 4.90-fold, respectively, compared to the control patients [16]. These data suggest that cadmium and lead may have a role in RCC initiation and/or progression.

Cadmium (10–30 μM; 9–15 weeks) induced EMT in lung adenocarcinoma cells [17]. Shaikh and co-workers have made significant progress in delineating a role for cadmium in breast cancer progression. MDA-MB-231 breast cancer cells challenged with low levels of cadmium (1–3 μM) for 8 weeks exhibited increased proliferation and migration, and EMT as evidenced by decreased E-cadherin expression and increased *n*-cadherin, Twist and slug expression [18]. These effects may involve wnt/β-catenin signaling and Snail expression [19,20]. Similar effects—decreased E-cadherin and increased *n*-cadherin expression—were also seen in MCF10A breast epithelial cells challenged with cadmium [20]. These in vitro studies are also supported by in vivo data. In the rat prostate, cadmium induced EMT as evidenced by loss of E-cadherin and increased vimentin [21]. In the normal kidney, cadmium induces markers of EMT [22,23], consistent with the loss of E-cadherin expression seen following cadmium challenge [24,25]. Neural cell adhesion molecules (NCAM) are a long-established target of lead [26,27,28] and more recent data indicate that lead may also alter *n*-cadherin expression [29]. Although a direct link between lead and EMT in tumor cells is not established, lead modulates the wnt/β-catenin pathway [30,31].

Given the association between cadmium or lead and RCC, and the impact of these metals on cell–cell adhesion, we hypothesized that cadmium and lead induced phenotypic changes reminiscent of EMT in RCC cells. In these studies, we investigated the impact of cadmium and lead on Renca cells, a mouse renal carcinoma cell line [32,33]. Renca cells, as opposed to human RCC cell lines such as 786-O cells, have the advantage of metastasizing following orthotopic injection into the kidney [33,34]. The impact of cadmium and lead on Renca cell cadherin and catenin expression, as well as the RCC phenotype, specifically aggregation, migration and invasion, was determined.

## 2. Results

### 2.1. Impact of Cadmium and Lead on Renca Cell Viability

Initial studies were designed to examine the impact of the environmentally relevant heavy metal contaminants, cadmium and lead, on RCC viability. Renca cells were challenged for 72 h (0–1.25 μM; 0.2% FB Essence) and viability assessed by the neutral red (NR), MTT and crystal violet assays. The highest concentration of cadmium and lead induced a significant loss of viability, as assessed by the MTT and crystal violet assays, while the lower concentration of cadmium and lead also decreased viability, as assessed by the crystal violet and MTT assay, respectively (Figure 1). Renca cells grow in a tight cluster of cells; when challenged with 1.25 μM cadmium or lead for 72 h, the number of larger clusters decreased and a spindle-shaped morphology was induced in some cells, most notably the cells not aggregated in the large cell clusters.

### 2.2. Impact of Cadmium and Lead on Cadherin and Catenin Expression

As the disruption of homotypic cell–cell adhesion is a fundamental step in EMT, we examined the impact of the metals on the cadherin/catenin complex using a similar challenge (0–1.25 μM; 72 h). In Renca cells, cadmium did not induce a substantial loss of E-cadherin (Figure 2). In fact, expression was increased at the low concentration, while lead induced a concentration-dependent loss of E-cadherin (Figure 3). Interestingly, cadmium, but not lead, induced a significant increase in p120-catenin expression at either concentration; this was shown to be an increase in isoform 1 using an isoform-specific antibody that recognizes isoforms 1 and 2, but not 3 and 4 [35] (Figure 2; Figure 3). Lead, but not cadmium, also induced a substantial loss of α-, β- and γ-catenin (data not shown). The data demonstrate that the two metals differentially affect the cadherin/catenin complex in Renca cells.

### 2.3. Impact of Cadmium and Lead on MMP-9 Expression

The overexpression of MMP-9 is linked to both EMT [36] and a poor prognosis in RCC [37,38]; therefore, we examined expression following metal challenge in Renca cells. Lead, but not cadmium (data not shown), induced a robust increase in MMP-9 expression (Figure 3).

### 2.4. Impact of Cadmium and Lead on Renca cell Adhesion, Migration and Invasion

To determine if the disrupted expression of the cadherin/catenin complex by cadmium and lead is consistent with decreased cell aggregation, the cells were challenged with cadmium and lead (1.25 μM) for 72 h and cell aggregation assessed for 2 h in the absence of metals. Both cadmium and lead significantly decreased the number of cell aggregates (10 cells or greater) (Figure 4). In a wound healing assay (scratch assay), both metals increased wound healing—cells were challenged with 1.25 μM metal for 72 h, and wound healing was assessed after 56 h in the absence of metals (Figure 4). Cell migration and invasion are hallmarks of metastasis; these parameters were assessed through Transwell assays using a serum gradient (0.25% to 5%) as the chemotactic gradient. Renca cells were challenged with cadmium and lead (1.25 μM) for 72 h, trypsinized and seeded onto the Transwell inserts in the absence of metals. Migration and invasion (Matrigel^®^ coated insert) were assessed at 48 and 72 h, respectively. Both metals increased migration and invasion (Figure 5). These data suggest that cadmium and lead induce phenotypic changes in RCC cells consistent with tumor progression.

## 3. Discussion

Taken together, the data indicate that cadmium and lead induce pro-metastatic alterations—i.e., decreased cell–cell aggregation, increased migration, and increased invasion—in established RCC cell lines. While the current study did not address the role of cadmium or lead in RCC initiation, they support the hypothesis that the metals are involved in RCC progression.

In high-grade RCC, sarcomatoid elements—characterized, in part, by the loss of the epithelial phenotype—are prevalent [39]; sarcomatoid RCC may be seen in any RCC subtype [40] and the presence of sarcomatoid elements is thought to be due to EMT [41]. There are four important steps involved in EMT: (1) the loss of epithelial cell adhesion; (2) actin reorganization/αSMA expression; (3) the disruption of the basement membrane; (4) enhanced cell migration and invasion [42]. Given the long association between cadmium and the disrupted expression and function of the E-cadherin/catenin cell adhesion complex [24,25], we examined cadherin/catenin expression in Renca cells following metal challenge. Surprisingly, the impact of cadmium on E-cadherin was minimal, while lead reduced E-cadherin expression in Renca cells. However, both metals were associated with the subsequent loss of cadherin/catenin function as assessed by cell aggregation. There were, however, metal-specific differences on cadherin/catenin expression. Lead, but not cadmium, elicited loss of α-, β- and γ-catenin expression, while cadmium, but not lead, increased p120-catenin isoform 1 expression. The impact on MMP-9 expression was also metal-specific, with lead, but not cadmium, inducing MMP-9 expression.

An interesting difference between cadmium and lead was the increased expression of the p120-catenin isoform 1 following cadmium challenge. p120-catenin (*CTNND1*) was first identified as an src tyrosine kinase substrate. Importantly, p120 phosphorylation correlated with src-induced transformation [43] and, not surprisingly, the phosphorylation of specific residues in p120-catenin is required for cell transformation [44]. The established role of p120-catenin in cancer is the subject of several reviews [45,46]. p120-catenin binds to the cadherin cytoplasmic domain, and shares sequence homology with β- and γ-catenin, but does not bind α-catenin [47]. The binding of p120-catenin to the cadherin intracellular domain stabilizes E-cadherin expression at the plasma membrane, whereas the absence of p120-catenin is associated with E-cadherin endocytosis and degradation [48].

Cloning p120-catenin revealed that there are four transcriptional start sites and four alternatively spliced exons, leading to multiple isoforms [49]. p120-catenin isoforms are named using the start site (1–4) and the alternatively spliced exons (A–D). While isoform-specific functions are largely unknown, the increased expression of p120-catenin isoform 1 is associated with cell invasion [50,51]. The increased expression of isoform 1 in motile cells, and upregulation in src-transformed MDCK cells, was one of the initial observations related to p120-catenin ([52]. In EMT, there is a transition from p120-catenin-3A to -1A splicing that parallels the E- to *n*-cadherin switch; the isoforms differ by 101-amino acids comprising the p120-catenin 1A *n*-terminus (head domain) [53]. In human epidermoid carcinoma and colon cancer cells, p120-catenin-1A enhanced invasion [50]. In RCC, p120-catenin gene expression is elevated approximately nine-fold in ccRCC patients [54] and, more importantly, there is a correlation between a switch to isoform 1A expression and RCC progression—89.2% of tumors that metastasized over the 5-year follow-up expressed p120-catenin isoform 1A [51]. The regulation of p120-catenin expression by cadmium represents an important future direction, and will give insight into a potential mechanism(s) of RCC progression.

In summary, the data provide compelling evidence for a role of cadmium and lead in RCC progression, specifically to a pro-metastatic phenotype. The molecular alterations correlate with phenotypic changes—increased wound healing, increased cell migration and invasion—that are associated with RCC metastasis. These data provide a strong rationale for future studies examining the role of lead in the regulation of cadherin/catenin and MMP-9 expression, and the role of cadmium in the E- to *n*-cadherin switch in RCC, as well as the aberrant expression of p120-catenin.

## 4. Materials and Methods 

### 4.1. Cell Lines

Renca (mouse; ATTC^®^ CRL-2947) cells were purchased from ATTC and cultured in RPMI1640 supplemented with 0.1 mM non-essential amino acids, 1 mM sodium pyruvate, 2 mM L-glutamine and pen-strep (50 U/mL and 50 mg/mL, respectively). Cells were used within the first 10 passages.

### 4.2. Cell Viability

Cell viability was assessed by the MTT assay which is based on mitochondrial conversion of soluble MTT to insoluble formazan. Renca cells were split 1:4 and seeded in a 96-well flat bottom tissue culture plate. After 24 h, cells were challenged with 0–1.25 μM cadmium chloride or lead acetate in media supplemented with 0.2% FB essence (VWR Cat#10803-034) for 72 h. Given the long half-life of cadmium in the body, renal concentrations of cadmium may reach millimolar concentrations [55] and the current lead blood reference level is 5 mg/dl (0.24 mmol/l). Two hours before harvest, 10 μL of 5 mg/mL MTT (Sigma Cat #M2128) dissolved in DPBS (Gibco Cat #14190-144), was added to each well. Upon harvesting, cells were washed with cold DPBS and dissolved by adding 50 μL solubilizing solution (10% Triton X-100, 0.1N HCl in isopropanol). Absorbance was measured using the 570/690 nm wavelengths on the Synergy HT Multi-Detection Microplate Reader (BioTek). Results are expressed by percent control as [Abs_570-690_ treated/Abs_570-690_ control × 100].

Cell viability was also estimated by the neutral red assay which stains lysosomes red in live cells. Renca cells were seeded on 96-well plates and challenged as described above. Three hours before harvest, 10 μL of 500 μg/mL neutral red (Sigma Cat #N4638), dissolved in 1X DPBS, was added to each well. After incubation, the media were aspirated and cells were fixed with 50 μL fixative solution (1% formaldehyde, 1% CaCl_2_) for 5 min. Fixative solution was aspirated, plates were dried in RT and dye was dissolved by adding 100 μL solubilizing solution (1% acetic acid, 50% ethanol) for 15 min. Absorbance was read at 540 nm on the Synergy HT Multi-Detection Microplate Reader. Results are expressed by percent control as [Abs_540_ treated/Abs_540_ control × 100].

The crystal violet method was also used for assessing cell viability [56]. Following 72 h challenge cadmium chloride or lead acetate as described, the media were aspirated, and the cells were stained for 10 min with 0.5% crystal violet. The cells were washed with water, and dye was extracted by the additional of methanol; absorbance was assessed at 570 nm and results expressed as percent control [Abs_570_ treated/Abs_570_ control × 100]. For visualization of cell morphology, cells were challenged in 10 cm dishes with metals for 72 h, and fixed for 10 min with 2% paraformaldehyde prior to staining with crystal violet (0.5%) for 10 min. After washing 3X with water, plates were dried and morphology assessed using an inverted Olympus IX51 microscope with the 10X objective (Olympus).

### 4.3. Western Blot

After the cadmium chloride or lead acetate challenge (0–1.25 μM for 72 h in media supplemented with 0.2% FB essence), cells were washed twice with ice-cold 1X DPBS and lysed with lysis buffer (10mM Tris-HCl, 1% SDS) containing Halt^TM^ Protease/Phosphatase inhibitors (Thermo Scientific Cat #78444). Cells were scraped and incubated on a rocker for 15 min at 4 °C. Cells were further disrupted by pipette 15 times and spun at 12,000 g_n_ for 15 min at 4 °C. Protein concentration was determined using the Pierce™ BCA protein assay kit (Thermo Scientific). Protein was separated on a 4–20% mini-PROTEAN^®^ gel (Bio-Rad) and transferred onto an Amersham™ Hybond™ PVDF membrane (GE Healthcare Life Sciences).

The following antibodies were used: anti-E-cadherin (BD Transduction Laboratories™ Cat #610182), anti-*n*-cadherin (BD Transduction Laboratories™ Cat #610920), anti-α-catenin [N1N3] (GeneTex Cat #GTX111168), anti-β-catenin (BD Transduction Laboratories™ Cat #610154), anti-γ-catenin (BD Transduction Laboratories™ Cat #610254), anti-p120-catenin (BD Transduction Laboratories™ Cat #610134), anti-p120-catenin isoform 1 and 2 (Sigma; clone 6H11), anti-MMP-9 (MBS153550, MyBiosource), and anti-β-actin (Sigma Cat #A2228); all primary antibody dilutions were 1:1000. Goat-anti-mouse HRP conjugate and Goat-anti-rabbit HRP conjugate (Jackson ImmunoResearch Laboratories, Cat #115035003 and 305035003) were used at 1:10,000 dilutions. Blots were developed using SuperSignal West Femto Chemiluminescent Substrate (Pierce Cat #34095) and imaged using the ChemiDoc^TM^ imaging system (Bio-Rad). Densitometric analysis was performed using Image Lab software V3.0 (Bio-Rad). The band volumes the for target proteins were divided by the b-actin value and then normalized to control values.

### 4.4. Cell Aggregation

Following a 72 h challenge with 1.25 μM cadmium chloride or lead acetate, cells were detached by incubating in 5 mL Moscona’s Low Bicarbonate Buffer (MLB) containing 2.5 mM EDTA for 10 min at RT and collected by scraping into a 15 mL tube. Plates were washed in 5 mL MLB, added to the tube and pelleted (1000 x g, 5 min). Cells were washed a second time with 5 mL MLB, counted, pelleted, and suspended to a final concentration of 5 × 10^5^ cells/mL in MLB/3 mM CaCl_2_/4 mM MgCl_2_. Twenty-four-well plates were pretreated with 500 µl MLB/1% BSA for 20 min and air-dried 20–30 min. One hundred µl of cells (5 x 10^4^) + 100 µl MLB/1% BSA were added to each well for a final concentration of MLB/1.5 mM CaCl_2_/2 mM MgCl_2_/0.5% BSA. In certain experiments, EDTA was added to a final concentration of 2 mM. Plates were incubated at 37 °C and visualized at 10x magnification at 2 h; clusters of >10 cells were counted.

### 4.5. Wound Healing

Cells were seeded in 12-well plates and dosed for 72 h with 1.25 μM cadmium chloride or lead acetate. A horizontal scratch was made using a 10 μL pipetman tip. Media were removed and the wells washed 2x with serum-free media. Cells were then incubated with 5% FB Essence for 56 h; cells were then fixed with 95% ethanol and stained with crystal violet. To quantify healing, the center area of the scratch was measured using the closed polygon tool (CellSense software).

### 4.6. Cell Migration and Invasion

Cell migration and invasion was assessed using the published methodology [57]. Following 72 h challenge with 1.25 μM cadmium chloride or lead acetate, Renca cells were split and seeded at 5 × 10^4^ cells/cm^2^ on Transwell plates (migration-8 μm pore PET membrane, Corning Cat#3464; invasion-8 μm pore PET membrane, Corning Cat#354480). The top chamber contained 0.2% FB essence; the bottom chamber contained 5%. After 48 h (migration) or 72 h (invasion), the top of the chamber was scraped with a cotton swab and inserts were fixed for 15 min in 70% EtOH. Cells were stained for 10 min in crystal violet, and cell migration and invasion were determined by manually counting the cells on the underside of the insert.

### 4.7. Statistics

Results are expressed as mean + standard deviation. A two-way analysis of variance (ANOVA) was performed, followed by Tukey’s multiple comparison test using the statistical software GraphPad Prism 8 (GraphPad Software, La Jolla, CA). The differences were considered statistically significant at *p* < 0.05.

## Figures and Tables

**Figure 1 ijms-20-06315-f001:**
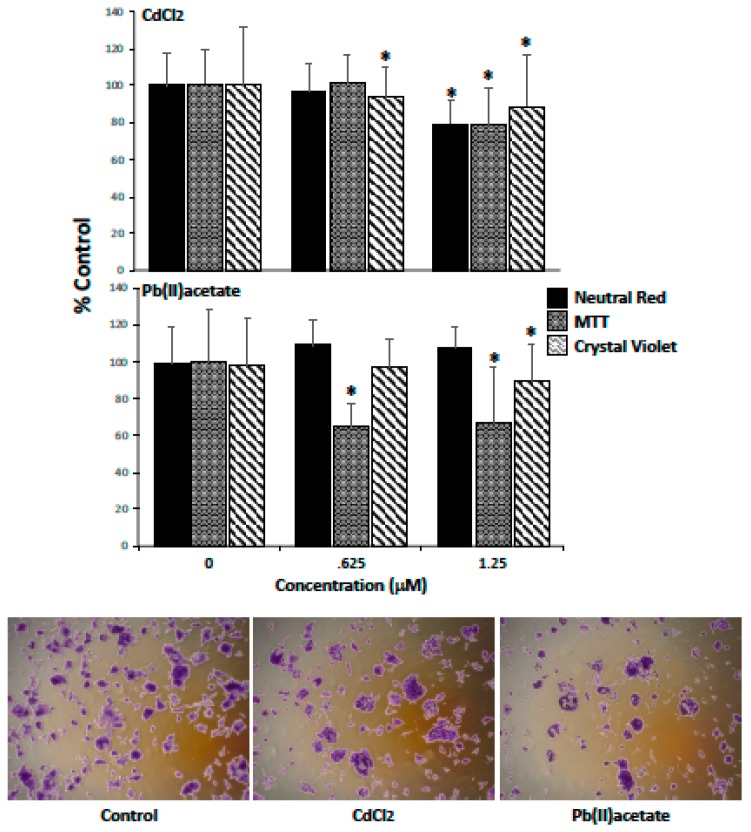
The impact of cadmium and lead on Renca cell viability. Renca cells were challenged with cadmium chloride or lead acetate (0–1.25 μM) for 72 h in media supplemented with 0.2% FB Essence. Cell viability was assessed with the neutral red, MTT and crystal violet assays. In the bottom panel, cell morphology was assessed by crystal violet staining after being challenged with 1.25 μM of cadmium or lead (10X objective) for 72 h. Each data point represents the mean + standard deviation of 70 replicates (% control) from three independent experiments; * indicates a significant difference from control; ^ indicates a significant difference from the low concentration (*p* < 0.05).

**Figure 2 ijms-20-06315-f002:**
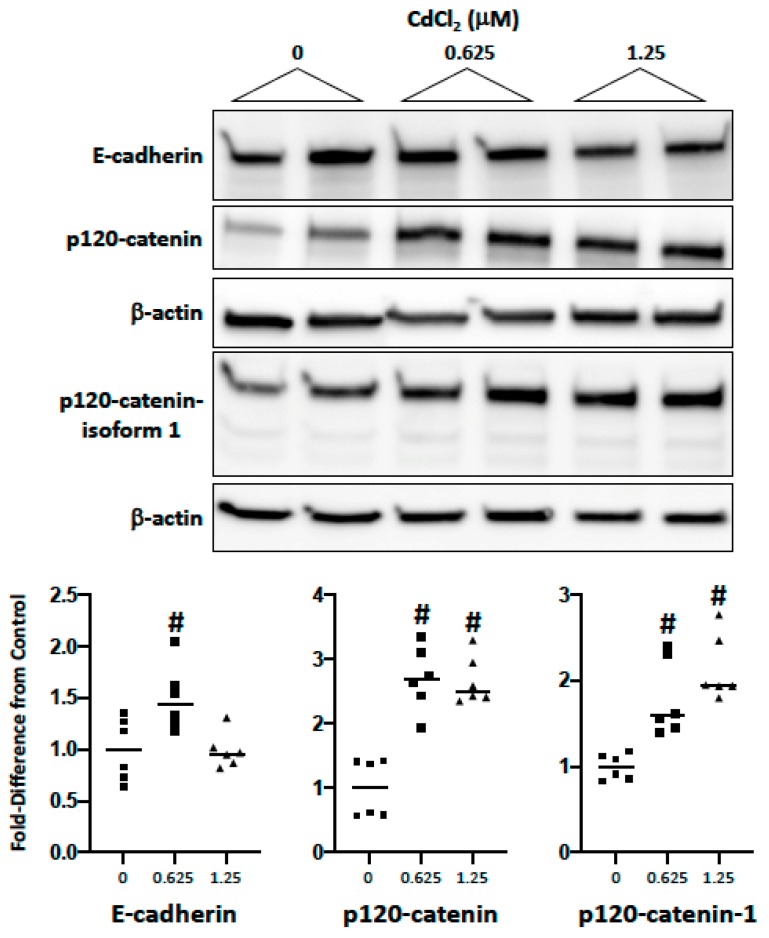
The impact of cadmium on E-cadherin and p120-catenin expression. Renca cells were challenged with cadmium chloride (0–1.25 μM) for 72 h in media supplemented with 0.2% FB Essence. Total cell lysates were harvested for Western blot analysis; similar results were seen in three independent experiments. In the bottom panel, the horizontal bars indicate the mean of six replicates from three independent experiments; # indicates a significant difference from control (*p* < 0.01).

**Figure 3 ijms-20-06315-f003:**
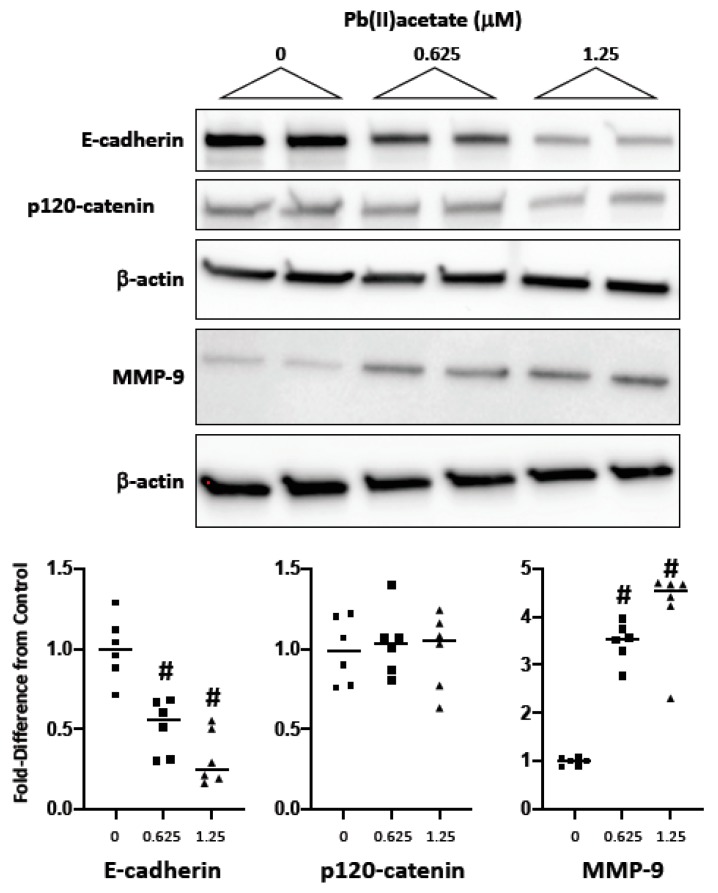
The impact of lead on E-cadherin, p120-catenin and MMP-9 expression. Renca cells were challenged with lead acetate (0–1.25 μM) for 72 h in media supplemented with 0.2% FB Essence. Total cell lysates were harvested for Western blot analysis; similar results were seen in three independent experiments. In the bottom panel, the horizontal bars indicate the mean of six replicates from three independent experiments; # indicates a significant difference from control (*p* < 0.01).

**Figure 4 ijms-20-06315-f004:**
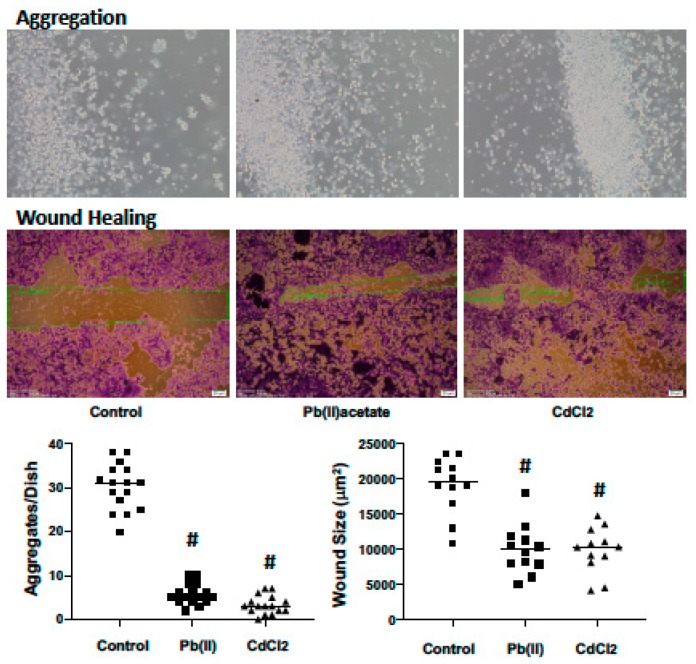
The impact of cadmium and lead on Renca cell aggregation and wound healing. Renca cells challenged with cadmium chloride or lead acetate (1.25 μM) in 0.2% FB Essence for 72 h. At this time, cells were lifted off the dish using Mosconas EDTA and aggregation was assessed at 2 h in Mosconas, or a wound was induced using a 10 μLpipet tip and healing assessed for 56 h in 5% FB Essence. Representative photomicrographs are shown in the top panel; in the bottom panel, the horizontal bars indicate the mean of 16 (aggregation) or 12 (wound healing) samples from two independent experiments; # indicates a significant difference from control (*p* < 0.01).

**Figure 5 ijms-20-06315-f005:**
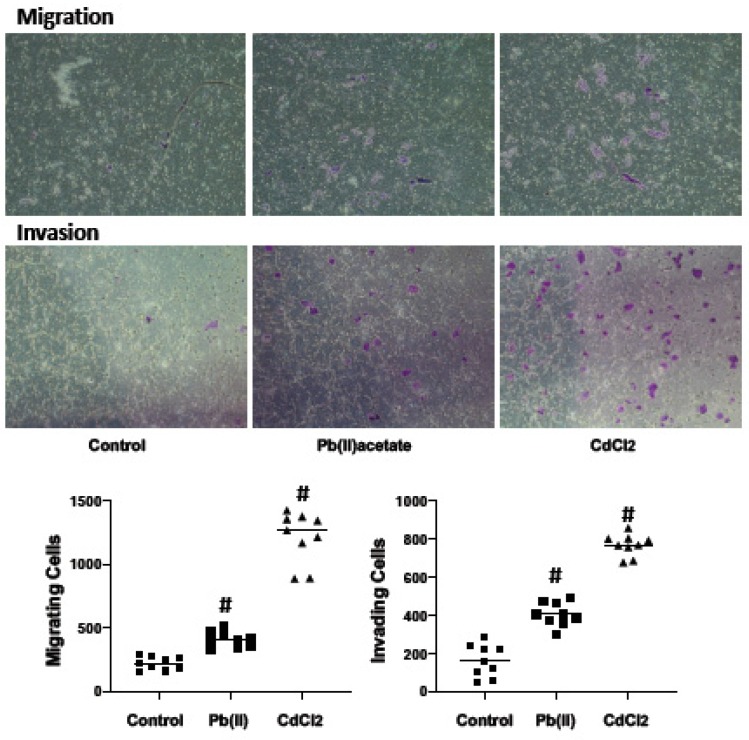
The impact of cadmium and lead on Renca cell migration and invasion. Renca cells challenged with cadmium chloride or lead acetate (1.25 μM) for 72 h in 0.2% FB Essence for 72 h. At this time, cells were trypsinzed and migration was assessed at 48 h in a Transwell plate or invasion (72 h in matrigel-coated Transwell plate) in response to a serum gradient. Representative photomicrographs are shown in the top panel; in the bottom panel, the horizontal bars indicate the mean of eight replicates from two independent experiments; # indicates a significant difference from control (*p* < 0.01).

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
