# Peer review of "Cadmium and Lead Decrease Cell–Cell Aggregation and Increase Migration and Invasion in Renca Mouse Renal Cell Carcinoma Cells"

_ijms, 2019, doi:10.3390/ijms20246315_

Round 1

Reviewer 1 Report

Despite numerous published studies, the possible role of heavy metals, and in particular of cadmium and lead in the genesis and / or progression of renal tumors, is still debated. A different chemical composition between the tumor tissue and the adjacent tissue has also been demonstrated. In this interesting experimental study the authors evaluate the impact of cadmium and lead on the cadherin/catenin complex in Renca cells, a mouse metastatic renal cell carcinoma cell line. The authors have shown alterations, linked to the presence of cadmium and lead, consistent with disruption of the cadherin / catenin complex and increased cell migration and invasion. Further studies will be needed to confirm the role of cadmium and lead in the progression of metastatic renal carcinoma.

Author Response

Thank you for the kind comments.

Reviewer 2 Report

This manuscript examines renal cell carcinoma using a mouse renal cell carcinoma cell culture model. The manuscript is based on the reported elevated plasma levels of lead and cadmium in individuals with renal cell carcinoma. Mechanistically, metals may alter cell to cell adhesion during the epithelial to mesenchymal transition.

GNERAL COMMENTS:

The authors must include in the revision the relationship between the concentrations of Cd and Pb sleected for exposure relative to what has been measured in humans exposed to these two metals in an acute situation.

The authors should further include the levels of plasma Pb and Cd measured in RCC individuals since the Introduction states the levels are elevated.

The results do not provide the statistical analysis of westerns. Consequently, the authors cannot make the conclusions stated in this manuscript until the results are presented in a complete manner.

SPECIFIC COMMENTS

Lines 35-37 Revise sentence as it addresses 2017 estimates in a present tense.

Lines 53-55 Provide a transition sentence for lead association with RCC.

Section 2.1 The authors need to comment on whether Pb or Cd alter cell viability prior ot 72 hours in the Renca renal cell carcinoma model. Measurements of cadherins and MMP are conducted at the time of loss of cell viability. Examining viability at earlier time points would be provide a characterization of cell death in the cell model.

Figure 1 The error bars for CdCl2 appear to overlap between groups with control for all test parameters, (neutral red, MTT and crystal violet). Can the authors check that the groups are statistically different, for 2 tailed tests, at 1.25 uM for all cytotoxicity tests as well as 0.625 uM (crystal violet)?

Figure 2 and 3 Missing Analysis of densitometry normalized to housekeeping protein B-actin and expressed as percent of control or comparison of each ratio. Statistical analysis should then be conducted to validate no statistical difference. Furthermore, Conclusions cannot be made regarding lead until statistical analysis is conducted. The conclusions by the author are not supported at this time by statistical analysis of densitometry.

Section 2.2 Provide the results for the α-, β- and γ- catenins for lead and cadmium. It is essential that the densitometry is provided as well as the statistical analysis.

Section 2.2 Does lead alter E-cadherin and MMP-9 prior to 72 hours?

Figure 4 Define the horizontal bars in the graph. Are these Mean or Median?

Figure 4 legend Revise to MEAN ± SD. Delete extra period at end of last sentence in figure legend.

Figure 4 were the aggregation and wound healing replicates or independent experiments? Figure legend states replicates which would suggest only an n=1. Were different passages used which would be independent studies? The conditions as to different experiments needs to be expanded.

Figure 5 legend to MEAN ± SD. Delete extra period at end of last sentence in figure legend.

Section 4.1 Provide the range of passages used for the studies conducted and reported in the manuscript.

Section 4.7 and Figures 1-4. The statistical analysis states a TWO WAY ANOVA was conducted followed by a TUKEY analysis. The figures and results only describe a difference back to control. Did the statistical analysis show an interaction for lead or cadmium? Address whether concentration dependent changes occurred from Pb in the western blots depicted in Figure 3.

Author Response

Reviewer 2

GNERAL COMMENTS:

Comments: The authors must include in the revision the relationship between the concentrations of Cd and Pb sleected for exposure relative to what has been measured in humans exposed to these two metals in an acute situation.

Reply: This information – estimation of renal cadmium levels in exposed populations and current blood lead standards – has been added in the Methods section.

Comments: The authors should further include the levels of plasma Pb and Cd measured in RCC individuals since the Introduction states the levels are elevated.

Reply:The values for metals in RCC patient blood, and lead levels in RCC tissue have been added in the Introduction.

Comments: The results do not provide the statistical analysis of westerns. Consequently, the authors cannot make the conclusions stated in this manuscript until the results are presented in a complete manner.

Reply:This has been added to the manuscript (below).

SPECIFIC COMMENTS

Comments: Lines 35-37 Revise sentence as it addresses 2017 estimates in a present tense.

Reply:We have corrected this sentence, thank you.

Comments: Lines 53-55 Provide a transition sentence for lead association with RCC.

Reply:A transition clause has been added to this sentence.

Comments: Section 2.1 The authors need to comment on whether Pb or Cd alter cell viability prior ot 72 hours in the Renca renal cell carcinoma model. Measurements of cadherins and MMP are conducted at the time of loss of cell viability. Examining viability at earlier time points would be provide a characterization of cell death in the cell model.

Reply:We have not examined any timepoints prior to 72 hr; we are extending the current studies  - which provide a rationale for studying the role of heavy metals in RCC progression - by using lower concentrations of metals in a subchronic dosing regimen. 

Comments: Figure 1 The error bars for CdCl2 appear to overlap between groups with control for all test parameters, (neutral red, MTT and crystal violet). Can the authors check that the groups are statistically different, for 2 tailed tests, at 1.25 uM for all cytotoxicity tests as well as 0.625 uM (crystal violet)?

Reply:The indications of significance on the graphs in Figure 1 are correct. 

Comments: Figure 2 and 3 Missing Analysis of densitometry normalized to housekeeping protein B-actin and expressed as percent of control or comparison of each ratio. Statistical analysis should then be conducted to validate no statistical difference. Furthermore, Conclusions cannot be made regarding lead until statistical analysis is conducted. The conclusions by the author are not supported at this time by statistical analysis of densitometry.

Reply:Densitometric analysis of the western blots is now included in the manuscript.

Comments: Section 2.2 Provide the results for the α-, β- and γ- catenins for lead and cadmium. It is essential that the densitometry is provided as well as the statistical analysis.

Reply:As cadmium did not have significant effects on catenin (other than p120 expression, shown in the paper), we have not included complete analysis of alpha, beta and gamma expression at this time; these experiments have only been repeated in duplicate.

Comments: Section 2.2 Does lead alter E-cadherin and MMP-9 prior to 72 hours?

Reply:We have not assessed any timepoints prior to 72 hr. 

Comments: Figure 4 Define the horizontal bars in the graph. Are these Mean or Median?

Reply:We have added the information that the horizontal bars represent the mean in the figure legends.

Comments: Figure 4 legend Revise to MEAN ± SD. Delete extra period at end of last sentence in figure legend.

Reply:Thank you for finding this error; it has been corrected.

Comments: Figure 4 were the aggregation and wound healing replicates or independent experiments? Figure legend states replicates which would suggest only an n=1. Were different passages used which would be independent studies? The conditions as to different experiments needs to be expanded.

Reply:The data in the graphs shown in Figures 4 and 5 are from two separate experiments; this information has been included in the Figure legend.

Comments: Figure 5 legend to MEAN ± SD. Delete extra period at end of last sentence in figure legend.

Reply:Thank you for finding this error; it has been corrected.

Comments: Section 4.1 Provide the range of passages used for the studies conducted and reported in the manuscript.

Reply:This information has been added in the Materials and Methods section.

Comments: Section 4.7 and Figures 1-4. The statistical analysis states a TWO WAY ANOVA was conducted followed by a TUKEY analysis. The figures and results only describe a difference back to control. Did the statistical analysis show an interaction for lead or cadmium? Address whether concentration dependent changes occurred from Pb in the western blots depicted in Figure 3.

Reply:Since the cadmium and lead groups used separate controls, we cannot make cadmium and lead comparison; however, we have added concentration-dependent changes to the Figures.

Reviewer 3 Report

This is a straightforward study examining the effects of minimally toxic concentrations of lead (Pb) or cadmium (Cd) on markers and phenotypes associated with cell migration and adhesion.  There are several issues which lowered this reviewer’s enthusiasm for this manuscript

Data in Figures 2 and 3 should be quantified before the authors can state that there were significant differences in protein expression. Additionally, if significant differences were observed, what are the p values? There is no information on the number of times each experiment was performed. Are the statistics based on technical replicates or biological replicates?  If the former, than biological replicates should be performed. Data for 786-O cells should be presented. If not they should be removed from the manuscript. The paragraph starting on line 185 is unnecessary, Line 96 p<0.5 or p<0.05

Author Response

Reviewer 3

This is a straightforward study examining the effects of minimally toxic concentrations of lead (Pb) or cadmium (Cd) on markers and phenotypes associated with cell migration and adhesion.  There are several issues which lowered this reviewer’s enthusiasm for this manuscript

Comments: Data in Figures 2 and 3 should be quantified before the authors can state that there were significant differences in protein expression. Additionally, if significant differences were observed, what are the p values?

Reply: Densitometric analysis of the western blots is now included in the manuscript.

Comments: There is no information on the number of times each experiment was performed. Are the statistics based on technical replicates or biological replicates?  If the former, than biological replicates should be performed.

Reply: The number of independent experiments performed has been added to each figure legend.

Comments: Data for 786-O cells should be presented. If not they should be removed from the manuscript.

Reply: The paragraph describing results in 786-O cells (data not shown) has been removed.

Comments: The paragraph starting on line 185 is unnecessary.

Reply: Given the impact of cadmium on p120 isoform 1 expression, we believe that this paragraph is necessary since this isoform is associated with RCC progression.

Comments: Line 96 p<0.5 or p<0.05 

Reply: Thank you for pointing out this error; it has been corrected.

Round 2

Reviewer 2 Report

The authors have addressed the reviewers' comments. 

Reviewer 3 Report

The authors have adequately responded of all of this reviewer's comments and concerns